# Hijacking Sexual Immuno-Privilege in GBM—An Immuno-Evasion Strategy

**DOI:** 10.3390/ijms222010983

**Published:** 2021-10-12

**Authors:** Martyn A. Sharpe, David S. Baskin, Amanda V. Jenson, Alexandra M. Baskin

**Affiliations:** 1Kenneth R. Peak Brain and Pituitary Tumor Treatment Center, Department of Neurosurgery, Houston Methodist Neurological Institute, Houston Methodist Hospital and Research Institute, Houston, TX 77030, USA; dbaskin@houstonmethodist.org (D.S.B.); avjenson@houstonmethodist.org (A.V.J.); ambaskin@houstonmethodist.org (A.M.B.); 2Department of Neurological Surgery, Weill Cornell Medical College, New York, NY 10065, USA

**Keywords:** GBM, HiF, Tregs, microglia, androgen, estrogen, sperm, testicular, myeloid-derived suppressor cells, tumor-associated macrophages

## Abstract

Regulatory T-cells (Tregs) are immunosuppressive T-cells, which arrest immune responses to ‘Self’ tissues. Some immunosuppressive Tregs that recognize seminal epitopes suppress immune responses to the proteins in semen, in both men and women. We postulated that GBMs express reproductive-associated proteins to manipulate reproductive Tregs and to gain immune privilege. We analyzed four GBM transcriptome databases representing ≈900 tumors for hypoxia-responsive Tregs, steroidogenic pathways, and sperm/testicular and placenta-specific genes, stratifying tumors by expression. In silico analysis suggested that the presence of reproductive-associated Tregs in GBM tumors was associated with worse patient outcomes. These tumors have an androgenic signature, express male-specific antigens, and attract reproductive-associated Related Orphan Receptor C (RORC)-Treg immunosuppressive cells. GBM patient sera were interrogated for the presence of anti-sperm/testicular antibodies, along with age-matched controls, utilizing monkey testicle sections. GBM patient serum contained anti-sperm/testicular antibodies at levels > six-fold that of controls. Myeloid-derived suppressor cells (MDSCs) and tumor-associated macrophages (TAMs) are associated with estrogenic tumors which appear to mimic placental tissue. We demonstrate that RORC-Tregs drive poor patient outcome, and Treg infiltration correlates strongly with androgen levels. Androgens support GBM expression of sperm/testicular proteins allowing Tregs from the patient’s reproductive system to infiltrate the tumor. In contrast, estrogen appears responsible for MDSC/TAM immunosuppression.

## 1. Introduction

Despite aggressive treatment, the median survival of GBM patients is only ≈15 months [1,2,3] and less than 5% survive ≥5 years postdiagnosis [1,4]. Swanson and colleagues’ computational tools estimate patient survival based on temporally distant T1-contrasted and T2-FLAIR images. They found long-term survivors of GBM (>36 months) tend to have slow-growing diffuse tumors with high T1-contrast and high FLAIR signal, whereas short-term survivors (<19 months) tend to have fast-growing focal tumors with little FLAIR signal [5,6]. Huang and colleagues recently demonstrated small hypoxic tumors display little FLAIR signal and correlate with poor outcome, confirming Swanson’s work [5,6,7]. The switch from aerobic to anaerobic respiration in cancers results from elevation of active hypoxia-inducible factor 1α (HiF) levels [8]. This activates transcription of hypoxia response genes, including Glut1 (*SLC2A1*) [9], Glut3 (*SLC2A3*) [10,11], *VEGFA* [12], *VLDLR* [13], and *ADM* [14]. Hypoxia, and the activation of HiF responsive genes in immune cells is well known to alter their function [15].

The response of the patient immune system to the GBM also affects patient outcome, with poor outcome being associated with tumor infiltration of myeloid-derived suppressor cells (MDSC) [16,17], tumor-associated macrophages (TAM) [18,19] and regulatory T-cells (Tregs) [20,21]. There may be a link between GBM tumor hypoxia and MDSCs and immunosuppressive macrophages. These are typically localized uterocervically, an area characterized by low glucose, low oxygen, low pH, and high lactate levels, much like a hypoxic tumor. Many of the components that induce uterocervical colonization by MDSCs/macrophages and their immunosuppressive phenotypes are also typically found in tumors, including GBM [22].

The levels of mRNA or protein markers of Treg infiltration into GBM tumors have been linked with poor outcome, including Forkhead Box P3 (*FOXP3*) [23], Cytotoxic T-Lymphocyte-Associated Protein 4 (*CTLA4*) [24,25], glucocorticoid-induced TNFR-related protein (*GITR*, (the *TNFRSF18* gene product)) [20,26] and GATA-binding protein 3 (*GATA3*) [27].

Th17-derived Tregs express the Related Orphan Receptor C (RORC) [28], along with FOXP3, CTLA4, and GITR [29]. These RORC-Tregs generally colonize mucin-rich tissues including in the gastrointestinal tract (GI) and kidney [29,30,31,32]. Additionally, the adult male and female reproductive tracts are lined with membrane-bound mucins and coated with soluble mucins, with levels of MUC1, 4 and 5AC elevated by androgen and estrogen. In these reproductive tissues the mucosal layer is infiltrated with Tregs [33,34].

MUC16 is a membrane-bound protein which generates a soluble mucin following proteolytic action, with the solubilized fragment sometimes called carcinoma antigen-125 (CA-125). Uterocervical CA-125 is generated in situ from MUC16 expressed in the cervix/uterus/endometrium [35] and high levels are an indicator of endometrial receptivity and a predictor of pregnancy [36]. Uterocervical CA-125 levels can be directly augmented from a sexual partner as seminal plasma has high levels of CA-125 [37]. This mucin fragment drives epitope-specific Treg infiltration and activation in normal reproductive biology and cancer, with CA-125 also expressed by ovarian tumors, generating an immunosuppressive niche [38]. The physiological function of the uterocervical RORC-Treg population is to aid the granting of immuno-privilege to ‘non-Self’ male-specific, partner-specific and offspring-specific seminal and gestational antigens [39,40,41]. In males, mucin expression within the reproductive tract begins at puberty, corresponding with their period of RORC-Treg colonization [42,43,44].

Most environments frequented by RORC-Tregs are hypoxic, with hypoxia aiding Th17 T-cells class switching into RORC-Tregs [45]. In the GI tract, typical *p*O_2_ ranges from 15.5 mmHg (resting) to 5.5 mmHg (digesting) [46,47]. The testicular *p*O_2_ is generally near 10 mmHg [48], prostatic *p*O_2_ near 5 mmHg [49], and in the vaginocervical mucosa, the *p*O_2_ is typically 1 mmHg [50,51,52]. These tissues which host RORC-Tregs are classifiable as hypoxic, defined as <15 mmHg, and indeed more hypoxic than a typical GBM tumor, which generally have a *p*O_2_ near 13 mmHg [53,54].

GBM U87 cells synthesize several steroids from cholesterol, including progesterone, androstenedione and dihydrotestosterone (DHT) [55,56]. Some primary GBM cultures express high levels of CYP17A1, allowing for efficient cholesterol utilization in steroidogenesis [57,58]. In addition to the canonical pathway of steroidogenesis, GBM may also use the ‘backdoor’ route, utilizing aldoketo reductases AKR1C1&3 [59].

The generation of sex steroids by GBM and other cancers initiates expressional changes in both cancer and infiltrating immune cells. Estrogen drives the epithelial to mesenchymal transition in glioma cultures and could act in the same manner in vivo [60]. Androgen receptor is elevated in both men and women’s GBM tumors and blockage of androgen signaling with enzalutamide in vitro leads to a loss of ‘stemness’, causing a reduction in cancer stem cell markers CD133, Nanog and Oct4 [61].

MDSCs and Tregs are antipodally modulated by estrogen and androgen. The *FOXP3* promoter region has an estrogen receptor α response element that controls expression [62,63], and estrogen suppresses Treg function and proliferation. In contrast, estrogen aids MDSC function, potentiated by progesterone [64]. Androgen elevates FOXP3 expression, acting through the androgen response element of the promoter [65,66]. Androgen also dampens the immunosuppressive functionality of MDSCs in vitro and in vivo [67].

We postulated that primary brain cancers, such as GBM, may initiate Treg infiltration by expressing sperm/testicular proteins, such as Glut14, which we have recently demonstrated in GBM primary cultures and in the GBM transcriptome database [68]. If a GBM were to present a sperm/testicular epitope to a reproductive RORC-Treg, then it would begin Treg infiltration and produce an immunosuppressive micro-environment.

Expression of sperm/testicular-specific proteins, some of which have been called cancer/testis antigens, has been observed in many cancer types, including GBM. Analysis of the immunopeptidome of GBM patient serum indicates tumor-derived testicular-, prostate-, placental- and fetal-specific antigens elicit an immune response in these patients regardless of gender [69]. Akiyama and co-workers identified several cancer/testicular antigens expressed by GBM tumors from cultured GBM cells [70], as did Freitas and colleagues [71]. Sreekanthreddy et al. identified pregnancy-specific glycoprotein 9 (PSG9) in GBM patient sera [72]. PSG9, like the other PSGs, activates TGFβ, which both lures and activates FOXP3^+^ Tregs [73], and also drives RORC-Th17^+^ cells from a proinflammatory to an immunotolerant form [74].

The sperm/testicular-specific gene *ACRBP* is expressed in GBM in different splice variants [75]. Li and co-workers examined both levels of ACRBP in glioma and the peripheral B-cell response to this protein in glioma patients. ACRBP mRNA transcripts were expressed in glioma and anti-ACRBP antibodies were found in patient sera [76].

The central question we are attempting to address is: what drives the accumulation and activation of immunosuppressive cells into GBM tumors?

We postulated GBMs evade immunosurveillance by two mechanisms. Firstly, they could engage in testicular mimicry, expressing sperm/testicular-specific and testicular-expressed proteins to both draw and activate reproductive Tregs pools keyed toward these sperm/testicular-specific epitopes. Men and women grant immuno-privilege to these epitopes via Tregs to counter immunological shock on the first encounter, and we briefly review this process in Appendix B. Secondly, the GBM tumors could emulate the environment of the placenta, expressing placental immunosuppressive-linked proteins, to provide a niche that attracts MDSCs and immunosuppressive macrophages. For instance, MUC1 is highly expressed in placenta [77], and expression is elevated by both estrogen [78] and progesterone [79]. Additionally, hypoxia elevates MUC1 expression [80], and MUC1 attracts and stimulates immunosuppressive MDSCs [81]. MUC1 is highly overexpressed in GBM and is negatively associated with overall survival [82].

**Markers of hypoxia response, immune cell levels, immune responses, androgen and estrogen reporters.** We have used a gene basket approach of tumor transcriptomes to examine the levels of cell types, status and for reporters of sex steroids. To stratify GBM into the proliferative, mesenchymal, and proneural subclasses, using the method of Teo et al. [83,84,85], we used the analysis performed by using the Gliovis platform [86].

Reportage of tumor HiF status was calculated from the averaged, normalized levels of hypoxia response genes: Glut1 (*SLC2A1*) [9], Glut3 (*SLC2A3*) [10,11], *ADM* [14], *VLDLR* [13], and *VEGFA* [12].

Reportage of tumor RORC-Treg infiltration was calculated from the averaged, normalized levels of *FOXP3* [23], *CTLA4* [24,87], *GITR* [20,26,88], *RORC* [29,45,89] and *GATA3* [27,90,91].

Reportage of tumor infiltration by microglia [MG] was calculated from the averaged, normalized levels of *ADORA3*, *IGSF6*, *TBXAS1*, *SASH3* and *P2RY13*. Inflammation status was calculated by dividing total MG by averaged, normalized levels of genes downregulated during inflammation, *P2RY12*, *TMEM119* and *GPR34*. Polarization by NF-κB was calculated by dividing averaged, normalized levels of genes known to be upregulated: *GCLC*, *NQO1*, *GCLM*, *NFKBIA* and *SLC39A8*, with averaged, normalized levels of genes known to be downregulated: *ZDHHC22*, *BCL7A* and *GNG4* [92].

Reportage of tumor infiltration by macrophages was calculated from the averaged, normalized levels of *ITGAM* and *CD68*. Polarization into M1 and M2 states was calculated by dividing total macrophage levels with *ITGAX* [M1] and with *CD163* (M2) transcript levels.

To estimate levels of androgen and estrogen in tumors, we used expression of androgen-controlled *KRT37*, and for estrogen we used the averaged, normalized levels of *THBD* [93,94], *THEMIS2* [95], *SERPINA1* [96], *PIK3CG* [97] and *VAV1* [98], respectively.

## 2. Results

### 2.1. HiF and RORC-Treg Gene Cross Correlation, Tumor Microenvironment and Impacts GBM Patient Outcome

#### Validation of HiF and RORC-Treg Gene Markers

There is a high degree of cross correlation between each of the five genes chosen to report HiF status across the four GBM transcriptome databases utilized.

In Appendix A, we show the cross-correlation plots, correlation coefficients and *p* values (from *t*-tests) of the five genes used for HiF reportage, with data taken from the Agilent (left) and Gravendeel (right) databases, respectively.

In Appendix A, we show the cross-correlation plots, correlation coefficients and *p* values (from *t*-tests) of the five genes we selected for RORC-Treg reportage, with data taken from the Aglient (left) and Gravendeel (right) databases, respectively.

These data clearly indicate that in these two databases, the genes chosen for the HiF and for the RORC-Treg are clearly correlated. Similar statistically significant correlations are found in the CGGA and U133 databases.

As RORC-Tregs of reproductive origin are normally associated with sperm/testicular-specific proteins and found in mucin rich tissues, we examined the cross-correlation plots, correlation coefficients and *p* values (from *t*-tests) of representative RORC-Treg-markers *CTLA4* and *RORC* with sperm/testicular-specific and mucin transcripts, Appendix A.

In Appendix A, we show that in GBM tumors, the expression of sperm/testicular-specific proteins, *ACRBP*, *SPATA12*, *TSSK6*, *HSPB9***,** and *CATSPER1* are highly cross-correlated with each other and with the representative RORC-Treg markers *CTLA4* and *RORC*. A similar pattern emerges when examining the cross-correlations of *CTLA4* and *RORC* with the mucin genes, *MUC5B*, *MUC6*, *MUC13*, *MUC16* and *MUC17*, Appendix A.

Appendix A, drawing on representative datasets Agilent and Gravendeel, show that RORC-Tregs can be present in GBM tumors, and that levels of RORC-Tregs are strongly and significantly correlated with sperm/testicular-specific proteins and mucins.

### 2.2. HiF and Tregs Impacts on Survival and Tumor Microenvironment

We examined the effect of RORC-Treg infiltration in hypoxic and normoxic tumors, by gating transcriptomes for previously described quintets of genes into high/low HiF and RORC-Treg. High/low cuts in each of the gene transcripts were made so as to obtain four roughly sized groups; Treg, Treg&HiF, HiF and neither. This was achieved by utilizing four databases: The Firehose Legacy Agilent microarray (201 patients), the CGGA database (180 patients) [99], the Gravendeel database (122 patients) [100] and the Firehose Legacy U133 platform (389 patients). This stratification methodology allows for assessment of both infiltrating Tregs (a property of the patient’s immune response) and tumor HiF status (derived from tumor metabolism and vascularization).

#### 2.2.1. Survival and HiF/RORC-Treg Status

Figure 1A shows four Kaplan–Meier survival curves stratified into Treg, Treg&HiF, HiF and neither categories. With each survival curve, the calculated *p* values and median survival (months) are indicated, along with dataset ID. We have omitted markers to identify censored patient endpoints (alive at time of archiving), but these data were used in the statistical analysis.

Patients whose tumors have infiltrating RORC-Treg or the HiF phenotype have a significantly worse outcome than normoxic and non-Treg-infiltrated tumors. Patients with Warburg phenotype tumors highly infiltrated by Tregs generally tend to have the worst outcome, with median survival <50% of the neither group.

We analyzed the reportage levels of HiF and RORC-treg gene baskets from all patients in the four databases, *n* = 892. Stratification by HiF/Treg results in the median HiF gene-basket reportage at levels 2.4 ± 0.7 times greater in the selected HiF subgroups than normoxic subgroups. The median RORC-Treg gene basket is expressed at levels 2.3 ± 0.9 times greater in the selected Treg subgroups than the negative pair; calculated *p*-values are *p* < 10^−10^ between pairs in both cases.

#### 2.2.2. CMP Subtypes

The distribution of CMP-subtypes in the stratified tumors are shown in Figure 1B, along with χ^2^ calculated *p*-values. In the hypoxic-Treg infiltrated tumors, the mesenchymal subtype dominates, in contrast to the normoxic-Treg infiltrated tumors with elevated classical and proneural sub-types.

#### 2.2.3. Stratification Reportage and Microglial Phenotypes

Levels of HiF and RORC-Treg markers of the four stratified groups are displayed in Figure 1C, using pie charts derived from 892 patients (U133, Aglient, Gravendeel and CGGA datasets). Total microglia levels were elevated in the neither subgroup, the only immunological cell type correlated with improved patient outcome, *p* < 0.01, Figure 1C. Poor patient outcome is correlated with microglia activation state with both proinflammatory and elevated NF-κB signaling over-represented in the at-risk subgroups.

#### 2.2.4. MDSCs

CD33 expression is a proxy for MDSC levels [101], and levels are overrepresented in hypoxic tumors, potentiated by RORC-Treg infiltration.

#### 2.2.5. Sperm/Testicular Proteins

Li and coworkers found antibodies toward sperm-specific protein ACRBP in the serum of their glioma patients [76]. We chose to examine *ACRBP* as a representative of sperm-specific gene expression in GBM tumors (see Appendix A). Expression of *ACRBP* is concordant with degree of Treg infiltration, but independent of HiF status, as shown in Figure 1C.

#### 2.2.6. Androgen and Estrogen Reporters and Macrophage Status

The expression of *KRT37* is controlled by three androgen response elements, and transcript levels can serve as an androgen/androgen receptor signaling proxy [102,103,104,105]. GBM tumor levels of *KRT37* are highly correlated with Treg infiltration. Appendix A shows the correlations between expression of *KRT37* (i.e., androgen expression) and the RORC-Treg gene basket, using the U133 dataset. Individual RORC-Treg markers are highly correlated with *KRT37*, all correlations *p* < 0.001, suggesting androgen-driven expansion and activation of Treg populations in GBM tumors [66]. In Figure 1D, we show pie charts representing the sex-steroid reporter levels and macrophage status using the U133 dataset. Using *KRT37* androgen reportage, it is clear that there is a strong correlation between androgen reportage and RORC-Tregs.

Our estrogen reporter is generated from a basket of normalized transcripts utilizing the quintet of estrogen-responsive genes: *THBD* [93,94], *THEMIS2* [95], *SERPINA1* [96], *PIK3CG* [97] and *VAV1* [98]. These gene expression levels are highly correlated, Appendix A, with all correlations *p* < 0.001. In Figure 1D, it is clear that estrogen reportage correlates with HiF status.

We calculated total macrophages levels using normalized levels of classical markers *ITGAM* and *CD68*. Hypoxia has been shown to aid macrophage recruitment and polarization toward the M2-phenotype [106], which is what we observe. The M2 > M1 hypoxic tumors are also noteworthy in their expression of anti-inflammatory, macrophage-associated, cytokine/chemokines IL10, EBI3 (co-protein in IL33) and CCL18.

In pursuit of a mechanistic explanation for immune responses in the subgroups, we examined the correlation of stratified subtypes with > 2200 genes. Figure 2 shows selected genes found highly expressed in the Treg, Treg&HiF and HiF subtypes (using the U133 dataset), *p* < 0.05.

The genes are clustered into groups based on cell type and function. The Treg subtype is to the left, HiF is on the right and Treg&HiF is centered, Figure 2. Bolded for emphasis are:(a)Keratin 37, our androgen reporter, correlates with Tregs.(b)Aromatase (*CYP19A1*), which converts testosterone to estrogen is elevated with HiF.(c)Sperm/testicular-specific gene transcripts *ACRBP*, *CATSPER1*, *LY6K*, *SPATA12* and *THEG* all correlate with Tregs.

### 2.3. Steroidogenic Status Impacts GBM Patient Outcome

#### 2.3.1. Steroidogenesis in GMB

An examination of steroidogenic enzyme gene levels in the stratified GBM tumors, Figure 1 and Figure 2, indicated that there were two pathways for steroidogenesis in GBM tumors: the canonical and ‘backdoor’ pathways. Based on this emergent property, we stratified the tumors from the four databases based on the expression of enzymes required for steroidogenesis. GBMs were first stratified by ≈50% based on the high expression of canonical steroidogenic pathway genes. Those with the canonical pathway were then subdivided, based on aromatase expression, into androgen- (with progesterone) and estrogen (with progesterone)-generating tumors. The remaining GBMs with low levels of the enzymes of the canonical pathway were then stratified into two equally sized fractions of those expressing the non-canonical ‘backdoor’ pathway genes, which generated dihydrotestosterone and estrogen, and the tumors which appear to be asteroidogenic (See Methods).

#### 2.3.2. Overall Survival and Status

Figure 3A shows the Kaplan–Meier survival curves of GBM patients stratified into steroidogenic phenotypes; androgen and progesterone, with low estrogen (A&P), estrogen and progesterone, with low androgen (E&P), estrogen and androgen, low progesterone (E&A), and asteroidogenic groups. Also shown is the median survival in months and the calculated *p* values with respect to the asteroidogenic group. Markers identifying censored patient endpoints were again omitted but used in statistical analysis. GBM steroidogenesis is highly detrimental to patient outcome, with median survival time of the E&A group only 75% that of the asteroidogenic group. Patients with A&P/E&P tumors fare worse, with only ≈60% of the longevity of the low steroid group.

#### 2.3.3. CMP Subtypes

In Figure 3B, we show the distribution of CMP subgroups, stratified by steroidogenic output, capturing the role of sex steroids in GBM etiology. Levels of mesenchymal cells in the three steroidal groups are consistent with reported effects of sex steroids on epithelial–mesenchymal transition. Reports state progesterone fetters the epithelial–mesenchymal transition [107], whereas androgen’s effects are much weaker [108]. In contrast, estrogen was shown to drive the epithelial–mesenchymal transition in many tissues, including GBM cells [60]. We find estrogen causes over-representation of mesenchymal phenotype GBMs, whereas progesterone favors the classical CMP subgroup.

#### 2.3.4. HiF and Tregs

The first two pie charts of Figure 3C show there is a small, statistically significant elevation in HiF reportage in tumors with the ability to generate androgen and estrogen, *p* < 0.05. Androgenic GBMs clearly show elevated levels of infiltrating RORC-Tregs, *p* < 0.001.

#### 2.3.5. Microglial Phenotypes

The two estrogenic subtypes induce microglial proliferation and have elevated NF-κB activation markers, which matches studies performed in vitro and in vivo [109,110]. In contrast, both progesterone and androgen suppress NF-κB pathway activation with progesterone acting directly through its receptor, but with androgen indirectly acting via cytokine signaling [111,112].

Microglia infiltration is elevated in estrogenic GBM, with levels elevated >40%, *p* < 0.001, and we observe >30% NF-κB signaling in estrogenic GBM, *p* < 0.001, Figure 3C. Unlike combinatorial estrogen and progesterone, combined androgen and progesterone has a synergistic effect on microglia NF-κB signaling, with the difference between A&P and E&P groups being 3:1, *p* < 0.001.

#### 2.3.6. MDSCs

GBM tumor infiltration by MDSCs is estrogen sensitive, with marker *CD33* overrepresented, *p* < 0.001, Figure 3C [113,114]. Elevated expression of the estrogen signaling reporter basket of genes correlates with MDSC and TAM marker genes [115], Appendix A, all correlations *p* < 0.001. This is consistent with our postulate that GBMs are steroidogenic, allowing the tumors to mimic reproductive tissues that attract either RORC-Tregs or MDSCs/TAMs.

#### 2.3.7. Sperm/Testicular Proteins

Expression of *ACRBP* is concordant with the degree of steroidogenesis, Figure 3C. Expression of *ACRBP* is strong in all three sex-steroid groups, all statistically significantly different from the asteroidogenic group, *p* < 0.001, but the two androgenic types are indistinguishable from each other. This observation supports the postulate that sperm-specific proteins are expressed by GBM to lure testicular/uterocervical Treg pools. From an analysis of the relative levels of different steroidogenic enzyme transcript levels, we present the routes and relative fluxes of steroidogenesis for the A&P, E&P and E&A groups in Appendix A. Additionally, the ability to block steroidogenesis with off-the-shelf clinical inhibitors is shown.

#### 2.3.8. GBM Steroidogenesis Validation, Macrophage Infiltration and Polarization

Estrogen acts as a potent anti-inflammatory regulator in healing and causes macrophage polarization M2 > M1 [116]. Progesterone acts in a similar manner to estrogen, driving M2 > M1 polarization [117]. The E&P group has the highest fraction of both total macrophages and M2-polarized macrophages. Androgens can initially aid M2 > M1 polarization; however, prolonged androgen exposure has been shown to cause M1 > M2 macrophage polarization [118], consistent with our stratification of GBM by steroidogenesis. Macrophage levels and polarization in the four types of tumors is consistent with the reported estrogen anti-inflammatory M2-phenotype, driving also the production of Il10 and IL33, but androgen driving M2 > M1 polarization. CCL18 generation by M2 macrophages is elevated by estrogen [119], and our data suggest that it is inhibited by androgen.

### 2.4. Elevation of Sperm/Testicular Specific Antibodies in GBM Patients

Emboldened by this in silico data, we reasoned that GBMs shed sperm/testicular-specific proteins into the peripheral lymphatic system. In adults, the levels of anti-sperm/testicular antibodies in circulation are very low, as reactive B-cell lineages are either deleted or converted into Bregs, either in testicles or by the seminized cervix (Appendix B). Shed sperm/testicular-specific antigens from a GBM tumor should generate anti-sperm/testicular-specific antibodies in patient serum. It has been demonstrated that following vasectomy or vaccination, sperm introduced in the periphery induce an immune response, and so high titers of anti-sperm antibodies develop [120,121,122,123,124,125]. Men, following vasectomy, have elevated circulating anti-sperm antibodies, but do not suffer from autoimmune orchitis because testicular immune privilege is maintained by Tregs [120,121,122,126].

We devised a methodology to interrogate GBM patient serum for multiple anti-sperm/testicular antibodies, whereby we labeled human antibodies with specificity toward proteins present in primate testicle slices. We used primate testicle as sperm-specific proteins generally have very high rates of evolution due to sperm competition, especially in small mammals [127]. This testicular tissue was obtained from control animals from a different study that had come to an end, and we were able to obtain and fix the tissue within minutes of euthanasia.

We incubated monkey testicular slices with serum from GBM patients and age-matched controls. Then, we labeled the tissue for the presence of human IgG. As a privileged organ, anti-sperm/testicular antibody binding to the monkey testicle should be low in both women and in unvasectomized men.

Figure 4 shows that the levels of anti-testicular antibodies were elevated in GBM patients compared with gallbladder surgery patients. In Figure 4A, we show a plot comparing the intensity of DAB-staining intensity in anti-sperm/testicular antibodies binding to monkey testicular tissue. We find nearly sevenfold greater levels of anti-sperm/testicular antibodies in GBM patient serum compared to control serum, *p* < 10^−10^.

Figure 4B,C allows the comparison of DAB (brown) labeling of monkey testicle with anti-sperm/testicular antibodies from the serum from two male and two female gallbladder surgery patients (B) and that from the serum of four male and four female GBM patients (C). The nuclei counter-stained blue.

The images in Figure 4B,C clearly show a heterogenous pattern of patient-derived antibody testicular labeling, which can be seen in the high magnification images, suggesting that different testicular epitopes are targeted/labeled in each patient’s serum. These antibodies arise from differential sperm/testicular-specific proteins expressed by an individual patients’ GBM, recruiting a separate pool of reproductive Tregs. The labeling of testicular slices with serum samples taken from other GBM and control patients are shown in Appendix A. We conclude two things: first, sperm/testicular-specific IgGs are present in GBM patient serum, and second, these antibodies arise due to sperm/testicular-specific proteins being expressed by the tumors.

## 3. Discussion

We have used a gene-basket approach to stratify GBM tumor transcriptomes, which has been used to examine the levels of specific cell types and signaling/synthetic path-ways in tumors. A drawback of this type of analysis is that although we can identify correlations between gene transcripts, we cannot be sure that the transcripts are present in the same cells. The tumor-infiltrating RORC-Tregs we have shown to be linked to poor patient outcomes and associated with tumor androgenesis were selected based on elevated levels of *FOXP3*, *CTLA4*, *GITR*, *RORC* and *GATA3* transcripts in the whole tumor, not from individual T-cells. A reasonable criticism of the in silico analysis performed is that these transcripts could be expressed by different cells within the tumor, and the correlations between the transcripts are spurious, in the same manner that rates of forest fires and ice-cream sales are spuriously correlated. With respect to the identification of tumor-infiltrating RORC-Tregs, additional evidence for this supposition is that the relationship between levels of the identified infiltrating RORC-Tregs and patient outcome is consistent utilizing data drawn from four different GBM transcriptome databases. Additionally, these RORC-Tregs transcripts are not only highly cross-correlated with each other, but also highly correlated with transcripts known to be recruiters of this Treg subclass, ‘male-specific’ *ACRBP*, *SPATA12*, *TSSK6*, *HSPB9* and *CATSPER1*, and with mucins *MUC5B*, *MUC6*, *MUC13*, *MUC16* and *MUC17*.

Our in silico analysis identified tumor steroidogenesis as a means for GBM to acquire an immunosuppressive phenotype, with the generation of androgens appearing to drive RORC-Treg infiltration and the generation of estrogen driving infiltration/activation of MDSCs/TAMs. Steroidogenesis can either be supported by the GBM cells, from non-cancer cells within the tumor, or an interplay between both. It has recently been reported that tumors can cause the expression of *CYP11A1* in T-cells, causing them to metabolize cholesterol to pregnenolone. This hormone is then released, resulting in immunosuppression [128]. The stratification of GBM tumors based on the synthesis of androgen and estrogen can be, and has been, validated by monitoring the transcripts of androgen-dependent (*KRT37*) and estrogen-dependent (*THBD*, *THEMIS2*, *SERPINA1*, *PIK3CG* and *VAV1*) reporter transcripts. We have presented a multiplicity of evidence for the generation of androgens by GBM tumors to be correlated with levels of RORC-Tregs, and for estrogen synthesis by these tumors to correlate with infiltration by immunosuppressive MDSCs and macrophages.

In GBM patients, we observe compartmentalization between an immune-privileged tumor microenvironment and a peripheral immune response, evident from the >6-fold increase in circulating anti-sperm/testicular antibodies. A similar discordant immune response is observed post-vasectomy. Vasectomy does not cause autoimmune orchitis, but the release of ‘male-specific’ proteins into the periphery causes elevated anti-sperm antibodies. Testicular Tregs maintain immune privilege following vasectomy, despite a peripheral immune response. This same immune compartmentalization appears to occur in GBM patients.

The heterogenous pattern of patient-derived antibody testicular labeling, which can be seen in the high magnification images, suggests different testicular epitopes are targeted/labeled in each patient’s serum. These antibodies arise from differential sperm/testicular-specific proteins expressed by an individual patients’ GBM tumors, recruiting a separate pool of reproductive Tregs. The reproductive Treg populations must be lured into the tumor microenvironment to become activated toward male-specific epitopes, and one of the most promising candidates is the mucins expressed by GBM. MUC16 has previously been shown to generate a Treg-attracting fragment after proteolysis, which has previously been shown to result in tumor aggressiveness in ovarian cancers [38].

## 4. Materials and Methods

The in silico transcriptome analysis we report is derived from four publicly available datasets using only IDH wild-type GBM tumors: the Firehose Legacy U133 (389 patients), the Firehose Legacy Agilent microarray (201 patients), the CGGA [99] database (180 patients) and the Gravendeel [100] database (122 patients). The classification of GBM sub-types refined by Wang et al. was taken from the analysis furnished by the Gliovis platform [86].

### 4.1. Stratification of GBM Phenotype by Warburg and Treg Infiltration Status

#### 4.1.1. Stratification of GBM Tumors by HiF

We used a basket approach to stratify tumors into positive/negative HiF based on expression of the hypoxia-response genes: Glut1 (*SLC2A1*) [9], Glut3 (*SLC2A3*) [10,11], *ADM* [14], *VLDLR* [13] and *VEGFA* [12]. Normalized mRNA expression levels of the five genes were rescaled between 0 (min) and 1 (max) and then averaged. The 5–10% fraction of tumors expressing low levels of each marker were relegated into HiF negative status, and the ≈50% tumors displaying the highest average expression were deemed as HiF positive. GBM tumor expression of HIF signature gene cross-correlations is shown in Appendix A using data obtained from the Firehose Legacy Agilent microarray [129] and the Gravendeel databases [100] with plots generated utilizing Gliovis [86]. *SLC2A1*, *SLC2A3*, *ADM*, *VLDLR* and *VEGFA* expression levels in GBM populations are highly cross-correlated with the other HiF markers, with *p* < 0.001 in all cases. To identify HiF status, we normalized and rescaled the expression levels of the quintet; the 5–10% fraction of tumors expressing low levels of each marker were deemed HiF negative; the ≈50% tumors displaying the highest average expression were deemed as HiF positive.

#### 4.1.2. Stratification of GBM Tumors by Tregs

After the transcriptiome data had been stratified into two similar groups representing hypoxic and normoxic, it was once more restratified. In the second round of stratification, the samples were divided on the basis of high/low levels of RORC-Treg markers. The GBM-infiltrating RORC-Tregs were selected on the basis of elevated expression of a 5-gene signature; *FOXP3* [23], *CTLA4* [24,87], *GITR* [20,26,88], *RORC* [29,45,89] and *GATA3* [27,90,91]. The cross-correlations of GBM tumor expression of these Treg signature genes is shown in Appendix A, with data obtained from the Firehose Legacy Agilent microarray and Gravendeel GBM tumor databases [100,129] and plots generated using the Gliovis platform [86]. The quintet of genes used to identify tumor infiltrating Tregs are highly cross-correlated with the other, with *p* < 0.001 in all cases.

To identify RORC-Treg high/low status, we normalized and rescaled the expression levels of the quintet; the 5–10% fraction of tumors expressing low levels of each marker were deemed RORC-Treg low, and the ≈50% tumors displaying the highest average expression were deemed as Treg high.

#### 4.1.3. Selection of Treg&HiF and Neither Groups

After assignment of HiF and Treg status, the data fell into four distinct groups: high Treg, high HiF, high Treg&HiF, and ’neither’ (low Treg and low HiF). The mRNA levels of highlighted gene expression levels with respect to the neither group is found in Appendix A (Excel Sheet, Treg&HiF Stratification), along with the fractional size of the four groups, for analysis from the four presented databases. The normalized average HiFs are 2.4 ± 0.7 times greater in the HiF-positive groups than in the HiF-negative groups and the Treg groups are 2.3 ± 0.9 times greater in the Treg high groups than in the Treg low groups across the four databases.

#### 4.1.4. Assaying Microglia Levels and Status

Given that microglia exhibit sex-specific phenotypes (Yanguas-Casás [130]), these immune cells were a major target of investigation. Tumor-infiltrating microglia and their inflammatory status were interrogated using the gene expression of 4 separate baskets, averaging the products of normalized/rescaled mRNA expression levels of each gene in a basket, with baskets drawn from transcriptome changes induced by LPS treatment or stimulation of NF-κB signaling in human microglial cultures (total MG: *ADORA3*, *IGSF6*, *TBXAS1*, *SASH3* and *P2RY13*; downregulated during inflammation, *P2RY12*, *TMEM119* and *GPR34*; upregulated by NF-κB: *GCLC*, *NQO1*, *GCLM*, *NFKBIA* and *SLC39A8*; and downregulated by NF-κB: *ZDHHC22*, *BCL7A* and *GNG4*) [92].

### 4.2. Stratification of GBM Phenotype by Steroidogenesis

#### 4.2.1. Choice of Genes for Stratification of Tumors by Steroidogenesis

We initially analyzed GBMs based on the genes of the canonical steroidogenesis pathway and planned to divide GBM tumors into androgenic, estrogenic and asteroidogenic groups. However, upon examining the outputs of androgen and estrogen reporter genes, it was evident that some GBMs had low transcript levels of critical canonical steroidogenic pathway genes, and yet appeared to be generating androgen and estrogen. This emergent property of the tumors led us to examine the non-canonic ‘backdoor’ steroidogenic pathway, which is utilized during gestation and in castration-resistant prostate cancers. We discovered that in some GBMs, this non-canonical pathway appears to be present, and so we used a stratification procedure that gave rise to four steroidogenic phenotypes: androgen (testosterone) with progesterone (A&P), estrogen with progesterone (E&P), estrogen with androgen (DHT) (E&A), and asteroidogenic.

#### 4.2.2. Canonical Steroidogenesis Groups A&P and E&P

The A&P and E&P subgroups use the canonical steroidogenesis pathways, expressing high levels of canonical pathway genes *CYP11A1*, *CYP17A1*, *HSD17B3* and *HSD3B1* alongside low levels of the progesterone catabolic enzyme *AKR1C1*. We selected the ≈50% of tumors with high levels of the first four genes and low *AKR1C1*. These were then stratified based on expression of aromatase (*CYP19A1*), which converts testosterone into estrogen, delineating the A&P and the E&P subtypes.

#### 4.2.3. Non-Canonical Steroidogenesis, the E&A Group

Approximately a quarter of the GBM tumors appear to employ non-canonic steroidogenesis, similar to estrogen-sensitive breast cancers [131,132] and castration-resistant prostate cancers [55,133,134,135,136]. Instead of beginning with cholesterol, steroid-sulfates are utilized, imported by fetal/placental transporters *SLC22A11*, *SLC22A19*, and *SLCO4A1*: the ‘side-entry’ synthetic route. These GBM then utilize *AKR1C1* and *AKR1C3* for ‘backdoor’ steroidogenesis [131,134,135,137,138,139,140]. Tumors lacking this non-canonical importation (asteroidogenic) were identified by low levels of all five gene transcripts.

#### 4.2.4. Selection of Androgen/Estrogen in the Four Groups

An emergent property was apparent during an initial analysis of the steroidogenic enzyme/transporter gene levels, indicating that GBM can be stratified into 4 groups: A&P (testosterone and progesterone), E&P (estrogen and progesterone), E&A (Estrogen and dihydrotestosterone), and ’neither’ (asteroidogenic). The mRNA levels with respect to the neither group are found in Appendix A (Excel Sheet, Sex-Steroid Stratification), along with the fractional size of the four groups for each of the four databases analyzed. Normalized average androgen reportage is 2.4 ± 0.7 times greater for two androgen-positive groups than in the asteroidogenic group, and estrogen reportage in the two estrogen groups is 2.0 ± 0.6 times greater than the asteroidogenic group.

#### 4.2.5. Visualization of Anti-Sperm Antibodies from Patient Sera

Unless otherwise stated, reagents were obtained from Sigma (Sigma-Aldrich, Inc., St. Louis, MO, USA). We probed primate testicle with patient serum, then visualized human antibodies with DAB staining. At necropsy, Cynomolgus monkey testicle was removed and fixed in 4% paraformaldehyde buffer for 7 days at 4 °C. The tissue was then washed and dehydrated using graded alcohol and xylene and then wax embedded. The block was sliced into 5 μm sections that were affixed to slides and dried. Slides were dewaxed four times in xylene, then twice in isopropanol, and rehydrated using graded ethanol. The slides were washed and permeabilized using Phosphate Buffered Saline (PBS, Fisher Scientific, Waltham, MA, USA) containing 0.1% Triton X-100. The slides were placed in Na-citrate buffer (100 mM, pH 6.0) and heated to <100 °C in a steamer for 30 min, followed by cooling slowly to bench temperature for epitope retrieval. After washing in PBS, endogenous peroxidase activity was eliminated using mild conditions: 1.8% H_2_O_2_ for 5 min, then in 1% sodium periodate for 5 min, followed by 0.02% NaBH_4_ for 2 min. Serum-Free Protein Block (Dako North America, Inc., Carpinteria, CA, USA) was applied for 1 h, and after washing, 50 μL of GBM or gallbladder surgery patient serum was applied and incubated at 4 °C overnight. After washing in PBS, 1:500 mouse anti-human IgG antibody (HP-6017 was applied and incubated for four hours. After washing, the HiDefTM HRP-polymer system (Cell Marque, Rocklin, CA, USA) was used to functionalize with peroxidase, and a DAB chromogen kit (Dako North America, Inc., Carpinteria, CA, USA) was used for visualization. Slices were then counter-stained using hematoxylin and sealed. Images were taken with a Carl Zeiss microscope (Oberkochen, Germany), and DAB labeling levels were quantified using ImageJ, as previously described [141,142]. We used an age-matched control group, as anti-sperm antibody titers have a slight age dependency [120,121]. Gallbladder surgery patient serum was used for control studies as these are demographically akin to GBM patients, and these samples were obtained from the Houston Methodist Biorepository. Anti-testicular antibodies were assayed using serum from GBM patients (10♀, 12♂) and gallbladder surgery controls (5♀, 6♂). All samples were treated simultaneously, in parallel, at each step of the immunohistochemistry process, incubated for the same time and using the same solutions.

## 5. Conclusions

Tumor hypoxia and infiltration by Tregs and also by CD33^+^ MDSCs are independent risk factors in GBM. Steroidogenesis is central to tumor infiltration by immunosuppressive cells. Estrogenicity creates a niche filled by MDSCs/TAMs. Androgenicity is correlated with expression of male-specific antigens and RORC-Treg infiltration and activation. These processes can potentially be halted with steroid inhibitors, improving patient out-come. Sperm/testicular-specific protein shedding by GBM can be assessed by examining levels of anti-sperm/testicular antibodies in the patient serum. These discoveries unlock novel methods for detection and aiding the treatment of GBM. The data herein strongly support the view that the main Treg subtype supporting GBM ferocity is the RORC-Treg class, a class of Tregs normally associated with mucin-rich cancers such as colorectal and pancreatic.

## Figures and Tables

**Figure 1 ijms-22-10983-f001:**
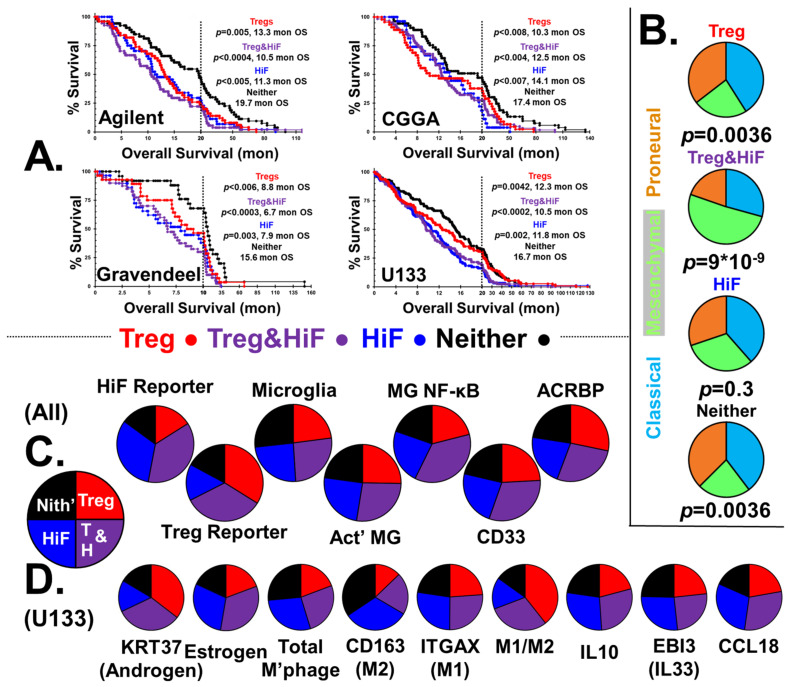
**GBM Patient infiltrating Tregs and HiF Status: Outcome and Characteristics.** (**A**) Color-coded Kaplan–Meier survival curves of GBM patients stratified for high Treg (red), high HiF (blue), high Treg&HiF (violet), and low Treg&HiF, known as ‘neither’ (black), Agilent, CGGA and Gravendeel and U133 databases. All *p* values are statistically significant, *p* < 0.05, when compared to the ‘neither’ group, and median overall survival is indicated in months. (**B**) The distribution of GBM subtypes in the four stratified groups, compared with the global distribution (37% classical, 32% mesenchymal, and 31% proneural), *n* = 892, as pie charts. Pie charts are color-coded with classical (light blue), mesenchymal (light green) and proneural (burnt orange). *p* values calculated from utilizing the χ^2^-test are shown. (**C**) Pie charts indicating seven markers of interest, *n* = 892, from normalized transcripts reporting medians. From left to right; HiF and Treg gene baskets validate stratification methodology. Total microglia (MG) are highest in normoxic tumors without Tregs, whereas both inflammation and NF-kB activation markers are elevated in HiF tumors. Myeloid cell marker *CD33* is elevated in hypoxic tumors. *ACRBP*, which cross-correlates with other sperm/testicular-specific proteins, correlates with RORC-Treg infiltration. *p* < 0.05 in all cases utilizing *t*-tests. (**D**) Data from U133 database, examining sex steroid and macrophage phenotypes in stratified groups in the form of pie charts. Androgen reportage (*KRT37* transcript levels) associates with RORC-Tregs, whereas hypoxia correlates with estrogen reportage. Macrophage infiltration (*ITGAM* and *CD68*) tracks hypoxia, and M2/M1 polarization markers *CD163* and *ITGAX* indicate RORC-Tregs appear to drive M1 > M2. *IL10*, *EBI3* and *CCL18* levels are biased toward hypoxic tumors. *p* < 0.02 in all cases utilizing *t*-tests. Individual *t*-test results in Appendix A.

**Figure 2 ijms-22-10983-f002:**
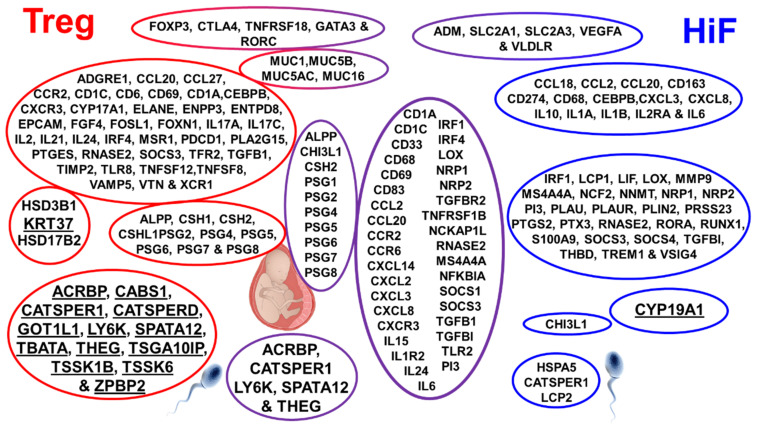
**Genes correlated with Treg and HiF clusters:** the breakdown of gene expression by high Treg expression subtype (left), high HiF expression subtype (right), and high Treg and HiF expression (middle). Genes clustered and highlighted are sperm/testicular proteins, including *ACRBP* and gestational genes such as PSGs. The androgen-responsive gene *KRT37* is associated with Tregs, as are many reproduction-associated genes, including sperm-specific, placental-associated and mucins genes. In contrast, aromatase (*CYP19A1*), which converts testosterone to estrogen, is associated with hypoxia. Noteworthy too is the HiF association with immunosuppressive M2 macrophage markers, including *CD68*, *CD69*, *CD163*, *MS4A4A*, and MDSC inducer *MUC1*. We also observe M2 macrophage cytokines *CCL2* and *CCL13* and *IL10* which can give rise to this polarization in hypoxic GBM.

**Figure 3 ijms-22-10983-f003:**
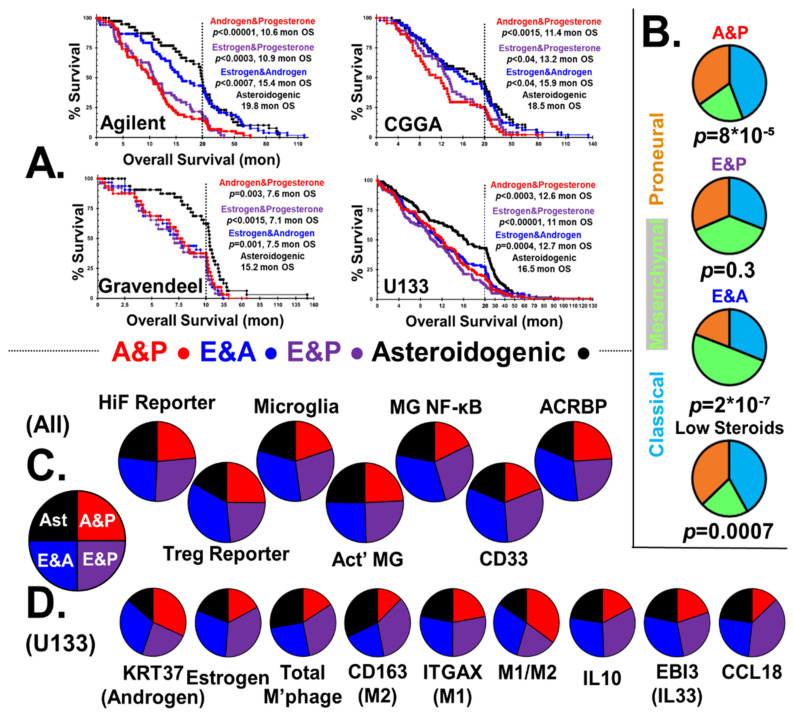
**GBM Patient Sex Steroid synthesis: Outcome and Characteristics.** (**A**) Kaplan–Meier survival curves of GBM patients with tumors stratified by high androgen/progesterone; low estrogen (red), high estrogen/progesterone; low androgen (violet), high estrogen/androgen production; low progesterone (blue), and overall low steroid production, asteroidogenic (black). All *p*-values are statistically significant compared to the low group. (**B**) CMP subtypes of GBMs correlate with androgen, estrogen and progesterone expression. (**C**) Data from all four databases in the form of pie charts indicating the amount of steroidal influence on seven markers of interest, all statistically significant at *p* < 0.05 except microglial activation. Estrogenic GBMs have elevated HiF and androgenic GBMs have infiltrating RORC-Treg reporter levels. Estrogenic GBMs are associated with microglial infiltration and NF-κB signaling. Myeloid cell infiltration reporter *CD33* tracks estrogen, with a 2:1 ratio of *CD33* levels in estrogenic tumors compared with tumors with low estrogen. Sperm-specific *ACRBP* clearly correlates with androgenic GBM subtypes. (**D**) U133 database pie charts indicating that androgen and estrogen reporter genes track the steroidogenic stratification. Total macrophage levels are low in androgen and progesterone. Sex steroids clearly affect the M2/M1 polarization, with androgen driving M1 > M2, but estrogen driving M2 > M1. Levels of immunosuppressive *IL10* and *CCL18* correlate with estrogen, but *EIB3* generation is inhibited by androgen. *p* < 0.02 in all cases utilizing *t*-tests. Individual *t*-test results are in Appendix A.

**Figure 4 ijms-22-10983-f004:**
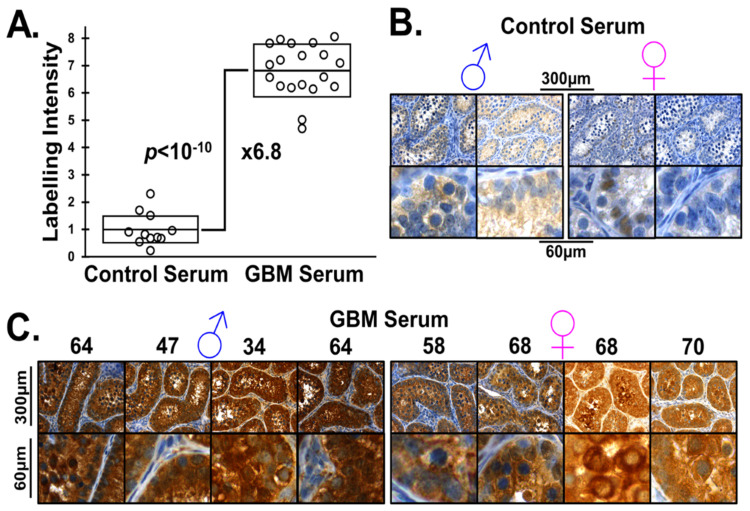
**Monkey Testicle labeled with Patient Serum.** (**A**) Staining intensity is significantly greater in the GBM than the control serum when brightness is quantified and scaled, nearly sevenfold that of the control, mean and SD boxed, *p* < 10^−10^. (**B**) Labeling of Cynomolgus monkey testicle using serum from control patients. Low levels of anti-sperm/testicular antibodies, and therefore light staining can be observed. (**C**) The significant staining on Cynomolgus monkey testicle when treated with patient sera indicates the presence of antibodies to testicular germ cells and to sperm at all stages of development. Patient age is atop the images.

## Data Availability

All transcriptome data used in the figures and in the Appendix A was obtained from the Firehose Legacy U133, the Firehose Legacy Agilent microarray, the CGGA and the Gravendeel databases, and references to these data are to be found within the text.The classification of GBM sub-types refined by utilizing the Gliovis platform, as were the graphics in some of the Appendix A. (http://gliovis.bioinfo.cnio.es/). Derived data is available on request from the corresponding author.

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
