# Peer review of "Hijacking Sexual Immuno-Privilege in GBM—An Immuno-Evasion Strategy"

_ijms, 2021, doi:10.3390/ijms222010983_

Round 1

Reviewer 1 Report

The manuscript by Sharpe and collaborators describes by using in silico analysis, the involvement of sexual proteins in stimulating an immunosuppressive environment in GBM. The paper is current, interesting and suggest a new way to explain the inhibition of the immune response in GBM.

However, a minor revision is required prior possible acceptance:

-Introduction is divided in sub-headings. Given that the manuscript is not a review, a continuous and more fluent style should be more suitable for the background.

-The methods used to investigate the presence of anti-sperm Abs should be improved. How did they evaluate the labelling intensity? What about animals? Why did they used monkey?

-Although the authors describe the vasectomy in the supplement text, they declare in the discussion in a very succinctly way that they used vasectomy as model. A clearer explanation and appropriate references should be added in this paragraph.

-According to authors, GBM by expressing sexual antigens recruit Tregs to promote an anti-inflammatory environment. This is interesting. According to the authors brain-infiltrated Treg are recruited for the presence of sexual proteins. Authors should better explain their hypothesis about the recruitment. Do they come from testicles? Are they only recruited for the presence of sexual proteins? Are additional chemotactic factors involved?

Author Response

Thank you for your thoughtful review of our paper.  We have considered each comment, and what follows is a point by point response 

 “Introduction is divided in sub-headings. Given that the manuscript is not a review, a continuous and more fluent style should be more suitable for the background”

We have removed the sub-headings and added linking sentences at the end of sections to improve fluidity of the paper.  We agree that this improves the readability of this section. 

“The methods used to investigate the presence of anti-sperm Abs should be improved. How did they evaluate the labelling intensity? What about animals? Why did they used monkey?”

We have expanded the methods section to fully describe the procedure: -

“4.2.5 Visualization of Anti-sperm Antibodies from Patient Sera. We probed primate testicle with patient serum, then visualized human antibodies with DAB staining. At necropsy Cynomolgus monkey testicle was removed and fixed in buffer 4% paraformaldehyde for 7 days at 4 °C. The tissue was then washed and dehydrated using graded alcohol, xylene and then wax embedded. The block was sliced into 5-μm sections that were affixed to slides and dried. Slides were dewaxed four times in xylene, then twice in isopropanol, and rehydrated using graded ethanol. The slides were washed and permeabilized using Phosphate Buffered Saline (PBS, Fisher Scientific, Waltham, MA) containing 0.1% Triton X-100. The slides were placed in Na-citrate buffer (100 mM, pH 6.0) and heated to <100°C in a steamer for 30 minutes, followed by cooling slowly to bench temperature for epitope retrieval. After washing in PBS, endogenous peroxidase activity was eliminated using mild conditions: 1.8% H2O2 for 5 minutes, then in 1% sodium periodate for 5 minutes, followed by 0.02% NaBH4 for 2 minutes. Serum-Free Protein Block (Dako North America, Inc., Carpinteria, CA, USA) was applied for 1 hour, and after washing 50 μl of GBM or gall bladder surgery patient serum was applied and incubated at 4 °C overnight. After washing in PBS, 1:500 mouse anti-Human IgG antibody (HP-6017, Sigma) was applied and incubated for four hours. After washing, the HiDefTM HRP-polymer system (Cell Marque, Rocklin, CA, USA) was used to functionalize with peroxidase, and the Dako DAB chromogen kit was used for visualization. Slices were then counter-stained using hematoxylin and then sealed. Images were taken with a Carl Zeiss microscope and DAB labeling levels were quantified using ImageJ, as previously described(141, 142). We used an age-matched control group, as anti-sperm antibody titers have a slight age dependency(120, 121). Gallbladder surgery patient serum was used for control studies as these are demographically akin to GBM patients, and these samples were obtained from the Houston Methodist Biorepository. Anti-testicular antibodies were assayed using serum from GBM patients (10♀, 12♂) and gall bladder surgery controls (5♀, 6♂). All samples were treated simultaneously, in parallel, at each step of the immunohistochemistry process, incubated for the same time and using the same solutions.

“Why did they used monkey?”

We have added text as to why we used primate testicle. We started using mouse testicle sections, but the amount of labeling we observed was poor. Many antibodies to sperm-specific proteins are highly species-specific, as sperm-specific proteins have high rates of evolution. Getting fresh, disease free human testicular tissue was not possible.

“We devised a methodology to interrogate GBM patient serum for multiple anti-sperm/testicular antibodies, whereby we labeled human-antibodies with specificity toward proteins present in primate testicle slices. We used primate testicle as sperm-specific proteins generally have very high rates of evolution due to sperm competition, especially in small mammals(127). This testicular tissue was obtained from control animals from a different study that had come to an end, and we were able to obtain and fix the tissue within minutes of euthanasia. 

We incubated monkey testicular slices with serum from GBM patients and age matched controls. Then, we labeled the tissue for the presence of human IgG. As a privileged organ, anti-sperm/testicular antibody binding to the monkey testicle should be low in both women and in unvasectomized men.”

-Although the authors describe the vasectomy in the supplement text, they declare in the discussion in a very succinctly way that they used vasectomy as model. A clearer explanation and appropriate references should be added in this paragraph.

We apologize for the unclear usage of the term ‘model’ in the first version. From the literature, it is known that following vasectomy, both men and mice have elevated levels of anti-sperm antibodies due to male-specific proteins leaking into the periphery. At the same time, the testicles maintain immune privilege and prevent autoimmune orchitis. This was our ‘model’ for circulating anti-sperm IgG’s present in patients even though they have an immunosuppressive tumor micro-environment. As we were ambiguous, we have rewritten the text.

“2.4 Elevation of sperm/testicular specific antibodies in GBM patients: Emboldened by this in silico data, we reasoned that GBMs shed sperm/testicular-specific proteins into the peripheral lymphatic system. In adults the levels of anti-sperm/testicular antibodies in circulation are very low, as reactive B-cell lineages are either deleted or converted into Bregs, either in testicle or by the seminized cervix (Appendix A). Shed sperm/testicular-specific antigens from a GBM tumor should generate anti-sperm/testicular-specific antibodies in patient serum. It has been demonstrated that following vasectomy or vaccination, sperm introduced in the periphery induce an immune response and so high titres of anti-sperm antibodies develop (120-125). Men, following vasectomy, have elevated circulating anti-sperm antibodies, but do not suffer from autoimmune orchitis because testicular immune privilege is maintained by Tregs(120-122, 126).”

“According to authors, GBM by expressing sexual antigens recruit Tregs to promote an anti-inflammatory environment. This is interesting. According to the authors brain-infiltrated Treg are recruited for the presence of sexual proteins. Authors should better explain their hypothesis about the recruitment. Do they come from testicles? Are they only recruited for the presence of sexual proteins? Are additional chemotactic factors involved?”

We have included a description of how MUC16 is proteolyzed into the soluble CA-125 fragment. CA-125 is known to cause Tregs to home into tumors and to activate an anti-inflammatory response. It predicts poor outcome in ovarian carcinoma because it induces an immunosuppressive environment.

MUC16 is a membrane bound protein which generates a soluble mucin following proteolytic action, with the solubilized fragment sometimes called carcinoma antigen-125 (CA-125). Uterocervical CA-125 is generated in situ from MUC16 expressed in the cervix/uterus/endometrium(35) and high levels are indicator of endometrial receptivity and predictor of pregnancy(36). Uterocervical CA-125 levels can be directly augmented from a sexual partner as seminal plasma has high levels of CA-125(37). This mucin fragment drives epitope-specific Treg infiltration and activation in normal reproductive biology and cancer, with CA-125 is also expressed by ovarian tumors, generating an immunosuppressive niche(38). The physiological function of the uterocervical RORC-Treg population is to aid the granting of immuno-privilege to ‘non-Self’ male-specific, partner-specific and offspring-specific seminal and gestational antigens(39-41). In males, mucin expression within the reproductive tract begins at puberty, corresponding with their period of RORC-Treg colonization(42-44).”

Reviewer 2 Report

This is a very complicated story to study immuno-evasion in GBM.

Introduction:

  1. Authors clearly introduce every key point in this study. However, the coherence of each point was not present. Hence, there is no rationales to conduct this study. (e.g. What is the correlation between hypoxia and sex steroids; immuno-microenvironment and sex steroids, Warburg effect and sex steroids....... )

Results

  1. Figures 1C-D and 3C-D are not clear. What do these 4 colors represent ??? Can these results analyzed statistically ??? How dose define "low steroids" in Figure 3C-D ??? Low steroids can not be represented by low expression of 1~2 steroidogenic enzymes. Authors may use another terms or organize results in the other way.
  2. I think that statistical significance is not observed in all pie charts. Dose Authors consider to re-analyze these results in the other way ??? 
  3. What is the connection between Treg, microglia, activated MG, Nf-kB, CD33 and ACRBP
  4. Why Authors studied ACRBP as a  sperm-specific gene ???
  5. Is any differences in steroidogenesis between hypoxic and normoxic GBM; between immune cell infiltration and non-infiltration ???
  6. In Figure 2, Authors analyzed genes correlated with Treg and HiF clusters . Whether these genes are involved in glioma progression ???
  7. The coherence between part 4-1 and 4-2 was not prepared well. Connection ???
  8. In Figure 4, " Labeling of Cynomolgus monkey testicle using serum from control patients". What did Authors label ??? ACRBP ??? 
  9. What is labeling intensity in serum in Figure 4A ??? What dose the signal labeled by serum means ??? Why did Authors label testicle ??? It is not easily understood. This protocol of serum preparation must also be clearly provided in the Materials section.
  10. If human specimens were used, please provided ethic number.
  11. Scheme 1: what is the role of sex steroid ??? Dose this talk about GBM ??? PGE2 and TGF-beta were not studied in any sections !!

Overall, I think this manuscript needs to be re-organized. Numerous information, too complicated. The take-home message is not clear. I even do not understand the purpose and significance of this study.

Author Response

We thank you for your review and your comments. In light of a careful reading of the many points made we have expanded the text. More importantly, we have changed the order of Figure/Supplementary Figures to reflect descriptions/discussions in text:-

Figure S1; Cross-correlation of selected hypoxia response genes HiF and RORC-Treg genes in GBM,

Figure S2; Cross-correlation of CTLA4 and RORC with male-specific and mucin genes in GBM,

Figure 1: GBM Patient infiltrating Tregs and HiF Status: Outcome and Characteristics.

Figure S3; Cross-correlation of RORC-Treg genes with androgen report gene KRT37, in GBM,

Figure S4; Cross-correlation of estrogen sensitive genes in GBM,

Figure 2. Genes correlated with Treg and HiF clusters:

Figure S5; Cross-correlation of MDSC/TAM markers and estrogen reporter basket in GBM,

Figure 3: GBM Patient Sex-Steroid synthesis: Outcome and Characteristics.

Figure S6; Steroid synthesis pathway with enzyme inhibitors,

Figure S7; Monkey testicle labeled with GBM and control patient serum,

Table 1; Characteristics of stratification of GBM tumors by HiF and Treg fractionation, with p values derived from t-tests performed against the neither cohort,

Table 2; Characteristics of stratification of GBM tumors by steroidogenic fractionation, with p values derived from t-tests performed against the asteriodogenic cohort.

This has greatly aided the narrative, and we have also added linker text between investigation blocks, which describe why and how we developed the investigation.

We have made updates to Figures 1 & 3. We have added a color-coded Key to make the interpretation of the pie charts easier. We have included the relative levels of androgen and estrogen reporters, which had previously only been included in the Supplemental tables and Figure 2. The inclusion of these hormone reporters in Figure 1D allows the easy visualization of the linkage of androgen levels with Tregs and the linkage of estrogen with HiF. This change, along with descriptions and call-outs to Figures S3 and S4, greatly strengthens the narrative.

Addressing your concerns in order:

“Authors clearly introduce every key point in this study. However, the coherence of each point was not present. Hence, there is no rationales to conduct this study. (e.g. What is the correlation between hypoxia and sex steroids; immuno-microenvironment and sex steroids, Warburg effect and sex steroids....... )

We agree that this can be more clearly explained.  We have added this link between Treg w/wo hypoxia in Figure 1 and the results of the investigation into sex-steroid synthesis in Figure 3:

“2.2.6 Androgen and Estrogen reporters and macrophage status: The expression of KRT37 is controlled by three androgen response elements and transcript levels can serve as an androgen/androgen receptor signaling proxy(102-105). GBM tumor levels of KRT37 are highly correlated with Treg infiltration. Supplemental Figure 3 shows the correlations between expression of KRT37 (i.e. androgen expression) and the RORC-Treg gene-basket, using the U133 dataset. Individual RORC-Treg markers are highly correlated with KRT37, all correlations p<0.001, suggesting androgen driven expansion and activation of Treg populations in GBM tumors(66). In Figure 1D, we show pie charts representing the sex-steroid reporter levels and macrophage status using the U133 dataset. Using KRT37 androgen reportage, it is clear that there is a strong correlation between androgen reportage and RORC-Tregs.

Our estrogen reporter is generated from a basket of normalized transcripts utilizing the quintet of estrogen responsive genes: THBD(93, 94), THEMIS2(95), SERPINA1(96), PIK3CG(97) and VAV1(98). These gene expression levels are highly correlated, Supplemental Figure 4, with all correlations p<0.001. In Figure 1D, it is clear that estrogen reportage correlates with HiF status.”

“Figures 1C-D and 3C-D are not clear. What do these 4 colors represent ??? Can these results analyzed statistically ??? How dose define "low steroids" in Figure 3C-D ??? Low steroids can not be represented by low expression of 1~2 steroidogenic enzymes. Authors may use another terms or organize results in the other way.”

We can see how they could be confusing. We have redrawn the figures to include a large key, showing the stratification and color.

The t-statistics are included in the text and figure legend.

The individual t-test results were included in Supplemental Table 1 and Supplemental Table 2. We have added a call-out to this information to the Figure legends “Individual t-test results in Supplemental Table 2.”

Instead of using the term ‘Low Steroids’, we now use ‘Asteroidogenic’

In Figure 3D, with the addition of androgen and estrogen reporter pie charts aids the reader, allowing them to observe that stratification of by the levels of steroidogenic enzymes shows very good concordance with the levels of androgen and estrogen reporters.

What is the connection between Treg, microglia, activated MG, Nf-kB, CD33 and ACRBP

We have expanded the text and make clear that we find tumor generation of androgen may drive expression of some sperm-specific proteins (ACRBP being an example), and that these, in combination with androgen, promote immune-suppression via Treg infiltration.

We also state that another route to immunosuppression is via estrogen generation, which attracts myeloid derived suppressor cells and macrophage infiltration, polarized into a M2 state. In the brain’s resident macrophages, the microglia, estrogen causes an upregulation of Nf-kB signaling.

Why Authors studied ACRBP as a  sperm-specific gene ???

We have updated the text to state why we chose ACRBP as an example of a sperm-specific protein

“2.2.5 Sperm/testicular proteins: Li and co-workers found antibodies toward sperm-specific protein ACRBP in the serum of their glioma patients(76). We chose to examine ACRBP as a representative of sperm-specific gene expression in GBM tumors (see Supplemental Figure 2A). Expression of ACRBP is concordant with degree of Treg infiltration, but independent of HiF status, as shown in Figure 1C.”

Is any differences in steroidogenesis between hypoxic and normoxic GBM; between immune cell infiltration and non-infiltration ???

The new versions of Figure 1D and Figure 3D now have androgen and estrogen reporter pie charts, helping the reader observe the stratification by Treg clusters with Androgen, whereas HiF is biased toward estrogen. Figure 3D now allows one to visually determine that the stratification based on the relative levels of steroidogenic enzymes is concordant with the expected levels of androgen and estrogen reporters.

“In Figure 2, Authors analyzed genes correlated with Treg and HiF clusters. Whether these genes are involved in glioma progression ???”

We have expanded the Figure legend to re-emphasize the correlation with HiF, estrogen synthesis via aromatase, and immunosuppressive M2 macrophages, reading as follows:

“Figure 2. Genes correlated with Treg and HiF clusters: The breakdown of gene expression by high Treg expression subtype (left), high HiF expression subtype (right), and high Treg and HiF expression (middle). Genes clustered and highlighted are sperm/testicular proteins, including ACRBP and gestational genes such as PSGs. The androgen responsive gene KRT37 is associated with Tregs, as are many reproduction-associated genes, including sperm-specific, placental-associated and mucins genes. In contrast, aromatase (CYP19A1), which converts testosterone to estrogen, is associated with hypoxia. Noteworthy too is the HiF association with immunosuppressive M2 macrophage markers, including CD68, CD69, CD163, MS4A4A, and MDSC inducer MUC1. We also observe M2 macrophage cytokines CCL2 and CCL13, and IL10 which can give rise to this polarization in hypoxic GBM.”

The coherence between part 4-1 and 4-2 was not prepared well. Connection ???

We agree that this connection is not obvious. We have now written an introduction into section 4.2, strengthening the connection between these two sections, reading as follows:

“4.2.1 Choice of genes for stratification of tumors by steroidogenesis.

We initially analyzed GBM based on the genes of the canonical steroidogenesis pathway, and planned to divide GBM tumors into androgenic, estrogenic and asteroidogenic groups. However, upon examining the outputs of androgen and estrogen reporter genes, it was evident that some GBM had low transcript levels of critical canonical steroidogenic pathway genes, and yet appeared to be generating androgen and estrogen. This emergent property of the tumors, led us to examine the non-canonic ‘backdoor’ steroidogenic pathway, which is utilized during gestation and in castration resistant prostate cancers. We discovered that in some GBM this non-canonical pathway appears to be present, and so we used a stratification procedure that gave rise to four steroidogenic phenotypes: androgen (testosterone) with progesterone (A&P), estrogen with progesterone (E&P), estrogen with androgen (DHT) (E&A), and asteroidogenic

In Figure 4, " Labeling of Cynomolgus monkey testicle using serum from control patients". What did Authors label ??? ACRBP ??? 

What is labeling intensity in serum in Figure 4A ??? What does the signal labeled by serum means ??? Why did Authors label testicle ??? It is not easily understood. This protocol of serum preparation must also be clearly provided in the Materials section.

We have added additional text to make it clear that we are assaying patient serum for any and all circulating antibodies that react with sperm/testicular antigens. We have emphasized that normally there are only very low levels of antibodies that can react with antigens from this privileged organ. We chose ACRBP as a representative sperm/testicular-specific protein in Figures 1 and 3.

“2.4 Elevation of sperm/testicular specific antibodies in GBM patients: Emboldened by this in silico data, we reasoned that GBMs shed sperm/testicular-specific proteins into the peripheral lymphatic system. In adults the levels of anti-sperm/testicular antibodies in circulation are very low, as reactive B-cell lineages are either deleted or converted into Bregs, either in testicle or by the seminized cervix (Appendix 1). Shed sperm/testicular-specific antigens from a GBM tumor should generate anti-sperm/testicular-specific antibodies in patient serum. It has been demonstrated that following vasectomy or vaccination, sperm introduced in the periphery induce an immune response and so high titres of anti-sperm antibodies develop (110-115). Men, following vasectomy, have circulating anti-sperm antibodies, but do not suffer from autoimmune orchitis because of Treg driven testicular immune privilege.

We devised a methodology to interrogate GBM patient serum for multiple anti-sperm/testicular antibodies, whereby we labeled human-antibodies with specificity toward proteins present in primate testicle slices. We used primate testicle as sperm-specific proteins generally have very high rates of evolution due to sperm competition, especially in small mammals(116). Secondly, using control animals from a different study that had come to an end, we were able to obtain and fix the tissue within minutes of euthanasia. 

We incubated monkey testicular slices with serum from GBM patients and age matched controls. Then, we labeled the tissue for the presence of human IgG. As a privileged organ, anti-sperm/testicular antibody binding to the monkey testicle should be low in both women and in unvasectomized men.”

In the discussion we have added:

“In GBM patients, we observe compartmentalization between an immune-privileged tumor microenvironment and a peripheral immune response, evident from the >6-fold increase in circulating anti-sperm/testicular antibodies. A similar discordant immune response is observed post-vasectomy. Vasectomy does not cause autoimmune orchitis, but the release of ‘male-specific’ proteins into the periphery causes elevated anti-sperm antibodies. Testicular Tregs maintain immune-privilege following vasectomy, despite a peripheral immune response. This same immune compartmentalization appears to occur in GBM patients.

The heterogenous pattern of patient-derived antibody testicular labeling, which can be seen in the high magnification images, suggests different testicular epitopes are targeted/labeled in each patient’s serum. These antibodies arise from differential sperm/testicular-specific proteins expressed by an individual patients’ GBM tumors, recruiting a separate pool of reproductive Tregs. The reproductive Treg populations must be lured into the tumor microenvironment to become activated toward male-specific epitopes, and one of the most promising candidates are the mucins expressed by GBM. MUC16 has previously been shown to generate a Treg attracting fragment after proteolysis, which has previously been shown to result in tumor aggressiveness in ovarian cancers(38)”

.

We have also expanded the methods section to give a more in-depth description of what was done:

“4.2.5 Visualization of Anti-sperm Antibodies from Patient Sera. We probed primate testicle with patient serum, then visualized human antibodies with DAB staining. At necropsy Cynomolgus monkey testicle was removed and fixed in buffer 4% paraformaldehyde for 7 days at 4 °C. The tissue was then washed and dehydrated using graded alcohol, xylene and then wax embedded. The block was sliced into 5-μm sections that were affixed to slides and dried. Slides were dewaxed four times in xylene, then twice in isopropanol, and rehydrated using graded ethanol. The slides were washed and permeabilized using Phosphate Buffered Saline (PBS, Fisher Scientific, Waltham, MA) containing 0.1% Triton X-100. The slides were placed in Na-citrate buffer (100 mM, pH 6.0) and heated to <100°C in a steamer for 30 minutes, followed by cooling slowly to bench temperature for epitope retrieval. After washing in PBS, endogenous peroxidase activity was eliminated using mild conditions: 1.8% H2O2 for 5 minutes, then in 1% sodium periodate for 5 minutes, followed by 0.02% NaBH4 for 2 minutes. Serum-Free Protein Block (Dako North America, Inc., Carpinteria, CA, USA) was applied for 1 hour, and after washing 50 μl of GBM or gall bladder surgery patient serum was applied and incubated at 4 °C overnight. After washing in PBS, 1:500 mouse anti-Human IgG antibody (HP-6017, Sigma) was applied and incubated for four hours. After washing, the HiDefTM HRP-polymer system (Cell Marque, Rocklin, CA, USA) was used to functionalize with peroxidase, and the Dako DAB chromogen kit was used for visualization. Slices were then counter-stained using hematoxylin and then sealed. Images were taken with a Carl Zeiss microscope and DAB labeling levels were quantified using ImageJ, as previously described(141, 142). We used an age-matched control group, as anti-sperm antibody titers have a slight age dependency(120, 121). Gallbladder surgery patient serum was used for control studies as these are demographically akin to GBM patients, and these samples were obtained from the Houston Methodist Biorepository. Anti-testicular antibodies were assayed using serum from GBM patients (10♀, 12♂) and gall bladder surgery controls (5♀, 6♂). All samples were treated simultaneously, in parallel, at each step of the immunohistochemistry process, incubated for the same time and using the same solutions.

If human specimens were used, please provided ethic number.

Institutional Review Board Statement: The study was conducted according to the guidelines of the Declaration of Helsinki and approved by the Institutional Review Board (or Ethics Commit-tee) of Houston Methodist Hospital (IRB# Pro00014547 and approval 09/2016).”

Scheme 1: what is the role of sex steroid ??? Dose this talk about GBM ??? PGE2 and TGF-beta were not studied in any sections !!

The mini-review we included describes how women generate Tregs and Bregs toward of male-specific proteins, under the influence of signaling molecules in semen, PGE2/TGF-B, and other molecules released by the cervix.

We do not have any evidence that GBM emulate the semenized cervix, to generate Tregs and Bregs in situ, but this may happen. We will show the in silico and immunohistology of PGE2 and other modulators, in GBM tumors, in the a future publication.

Overall, I think this manuscript needs to be re-organized. Numerous information, too complicated. The take-home message is not clear. I even do not understand the purpose and significance of this study.

We have followed your advice and reordered the manuscript and expanded the text, with more callouts in the text.

We appreciate the careful and very detailed and focused advice you have given.  We believe that by doing this the manuscript is much improved, and hope it is acceptable for publication. 

Round 2

Reviewer 2 Report

It is appreciated that Authors made numerous efforts in responding my concerns point by point. I believe that sexual difference in GBM pathogenesis will be highly focused in the future. This study is a pioneer study in this field.

I have no further questions about this manuscript.

I recommend to accept it.